Effects of perioperative massive transfusion on postoperative outcomes of children undergoing brain tumor removal: a retrospective cohort study

Xu Yingyi 1 2
Zhang Na 2
Ou Xinxu 2
Ye Yuyin 3
Liu Jianhua 2
Zhang Siyi 2
Xu Xinke 4
Gao Yu 2
Chen Wenchu 2
Song Xingrong 1 2 sxjess@126.com
1 The First Affliated Hospital of Jinan University , Guangzhou , China
2 Department of Anesthesiology, Women and Children’s Medical Center, Guangzhou Medical University , Guangzhou , China
3 School of Pediatrics, Guangzhou Medical University , Guangzhou , China
4 Department of Neurosurgery, Women and Children’s Medical Center, Guangzhou Medical University , Guangzhou , China
Guan Fanglin
Electronic publication date: 2025 May 13
Publication date: 2025
Volume: 13
Electronic Location ID: e19350
Received 2025 Feb 4; Accepted 2025 Mar 28
Copyright: © 2025 Xu et al.
Copyright year: 2025
Copyright holder: Xu et al.
License: This is an open access article distributed under the terms of the Creative Commons Attribution License, which permits unrestricted use, distribution, reproduction and adaptation in any medium and for any purpose provided that it is properly attributed. For attribution, the original author(s), title, publication source (PeerJ) and either DOI or URL of the article must be cited.
License URL: https://creativecommons.org/licenses/by/4.0/

Keywords: Pediatric patients, Brain tumor surgery, Extensive blood transfusion, Clinical outcomes, Postoperative complications

Funding: Guangzhou Science and Technology Projects Key Research and Development program 202206010006 Guangzhou Health Technology Programme 20231A011032 Guangzhou Municipal Health Commission, General Guidance program 20241A011034 This project was supported by Guangzhou Science and Technology Projects Key Research and Development program (No. 202206010006, Yingyi Xu) and the Guangzhou Health Technology Programme (No. 20231A011032, Xinxu Ou), as well as the Guangzhou Municipal Health Commission, General Guidance program (No. 20241A011034, Jianhua Liu). The funders had no role in study design, data collection and analysis, decision to publish, or preparation of the manuscript.

==============================
Objective

To examine the influence of massive perioperative transfusion on both short-term and long-term outcomes in children undergoing brain tumor resection.

Methods

This retrospective cohort study analyzed pediatric patients who underwent brain tumor surgeries at the Women and Children’s Medical Center, Guangzhou Medical University, between October 2014 and March 2022. Massive transfusion was characterized as the transfusion of red blood cells equivalent to or exceeding the estimated preoperative blood volume within 48 h after surgery. To evaluate the relationship between massive transfusion and patient outcomes, logistic regression models were utilized. Survival curves were constructed to compare the long-term outcomes of patients who received massive transfusion with those who did not. The primary outcomes assessed were 30-day all-cause mortality (short-term outcomes) and 1-year survival rates (short-term outcomes).

Results

Among the 306 patients included in the analysis, 78 were categorized as part of the massive transfusion group, while 228 were in the non-massive transfusion group. Multivariable regression analysis indicated that perioperative massive transfusion was significantly linked to an increased risk of 30-day mortality (odds ratio (OR): 0.137, 95% confidence interval (CI) [0.036–0.528], p = 0.004). Additionally, patients in the massive transfusion group exhibited higher incidences of postoperative intracranial hypertension (OR: 4.788, 95% CI [1.547–14.824], p = 0.007), extended mechanical ventilation duration (OR: 0.247, 95% CI [58.739–147.895], p < 0.001), and prolonged intensive care unit (ICU) stays (ß = 0.184, 95% CI [3.874–15.077], p = 0.001).

Conclusions

Massive transfusion has a pronounced impact on short-term outcomes, particularly increasing perioperative mortality and complication risks in children undergoing brain tumor surgery. Careful consideration of the risks and benefits of transfusion is crucial in managing these cases.

Introduction

Tumors of the central nervous system rank as the second most prevalent malignancy in children, with an estimated incidence of 2.0 per 100,000, accounting for approximately one-third of all cancer cases in individuals under 16 years old (Helligsoe et al., 2022). Advances in neuroimaging, surgical techniques, neuro-oncology, and pediatric intensive care have facilitated more aggressive tumor resection strategies (Hodžić et al., 2020). However, surgical removal of brain tumors, particularly via craniotomy, carries a substantial risk of significant blood loss and the necessity for transfusion (Keung et al., 2009). This is largely because many of these tumors are malignant and predominantly arise in the posterior fossa (Piastra et al., 2008).

Intraoperative massive hemorrhage arises from multiple factors, including the intricate nature of brain surgery, the highly vascularized brain anatomy, and challenges in achieving hemostasis, which are often exacerbated by limited surgical expertise (Vassal et al., 2016). In cases of excessive intraoperative blood loss, allogeneic transfusion can replenish the blood volume of pediatric patients, maintain stable hemodynamics, and rapidly increase the hemoglobin level of patients, thereby improving tissue oxygen supply. However, the use of massive allogeneic blood transfusions in pediatric patients also poses considerable risks, including coagulopathy, transfusion-related acute lung injury, acute kidney injury, and infections (Catarci et al., 2023; Glance et al., 2011). Moreover, immunosuppression induced by transfusion may impair tumor surveillance mechanisms, potentially leading to unfavorable outcomes (Muszynski et al., 2017). Despite these risks, the effects of massive allogeneic transfusion during brain tumor surgery on postoperative long-term outcomes remain poorly understood, particularly in large pediatric cohorts. To address this gap, we conducted a retrospective cohort study to compare postoperative outcomes between pediatric brain tumor patients who underwent massive transfusion and those who did not.

Materials and Methods

Study design

This retrospective analysis was conducted on pediatric patients who underwent brain tumor removal at the Guangzhou Women and Children’s Medical Center, Guangzhou Medical University, between October 2014 and March 2022. Data were obtained from retrospective review of our institution’s electronic medical record with the access period starting at July 5, 2024 and ending at July 7, 2024. Ethical approval was obtained from the Research Ethics Committee of Guangzhou Women and Children’s Medical Center (Approval No. 2024239A01, dated July 1, 2024). All procedures adhered to the principles outlined in the Declaration of Helsinki. Written informed consent was obtained from patients’ legal guardians, allowing their data to be used for future research purposes.

Patients under 18 years of age who were admitted for intracranial tumor resection and whose diagnoses were confirmed by postoperative histopathological examination were eligible for inclusion. Exclusion criteria encompassed incomplete medical records, multiple or recurrent tumors, preoperative blood transfusion history, or the absence of intraoperative transfusion.

Hematological assessment

Massive allogeneic transfusion was defined as red blood cell (RBC) transfusion equal to or exceeding the preoperative estimated blood volume within 48 h following surgery. Transfused blood volume was calculated using the formula: EstimatedRedCellVolumeTransfused(ERCT)=PBRC/0.6. Here, PBRC refers to the volume of packed red blood cells transfused, with a hematocrit of 0.6 as provided by the Guangzhou Central Blood Station. The estimated total blood volume was 80 ml/kg for infants below 2 years and 70 ml/kg for children aged 2 to 18 years.

Definitions of variables and data collection

The study examined the impact of perioperative allogeneic transfusion on clinical outcomes. Data were retrieved from the hospital’s electronic medical record system, focusing on 20 potential confounding variables. These included 13 preoperative patient-specific characteristics, such as age, sex, weight, external ventricular drainage history, American Society of Anesthesiologists (ASA) physical status, Karnofsky Performance Scale (KPS) score, anemia (defined as hemoglobin <110 g/L in the last preoperative test), and coagulopathy (platelet count <100,000/mm3, international normalized ratio >1.2, or activated partial thromboplastin time >36 s) (Talving et al., 2011).

Tumor-related variables included edema, location (supratentorial or subtentorial), number (single or multiple), brain invasion, and tumor size. Tumor characteristics were assessed using magnetic resonance imaging (MRI), with brain invasion defined as infiltration into nearby structures such as the skull base, dura mater, arachnoid, brain, or brainstem. Tumor size was measured as the largest axial diameter.

Intraoperative parameters encompassed surgical duration, blood loss, fluid infusion, and the extent of tumor resection. Resection extent was categorized using a five-point brain tumor resection score: (1) extended tumor removal; (2) complete tumor removal; (3) complete intracranial tumor removal with preservation in critical nerve functional areas; (4) incomplete tumor removal; (5) biopsy or decompression without tumor removal (Baumgarten et al., 2016). Postoperative data included Ki67 proliferation index and tumor grading based on the current World Health Organization (WHO) classification of central nervous system tumors (Louis et al., 2016).

Outcomes and follow-up

The primary outcomes of this study were 30-day mortality and 1-year survival rate. Among these, 30-day mortality was classified as a short-term outcome, while one-year survival rate represented the long-term outcomes. Children were monitored from their initial hospital admission until March 1, 2023, or the date of death, whichever occurred first.

The study also evaluated 10 secondary outcomes. Three of these focused on postoperative complications: infections (including intracranial infections, surgical site infections, pneumonia, and bloodstream infections), re-craniotomy due to bleeding or swelling, and intracranial hypertension. Additional secondary outcomes included the duration of mechanical ventilation, length of stay in the intensive care unit (ICU) and hospital, new neurological events (Table S1), and tumor recurrence or metastasis.

The diagnosis of postoperative neurological and non-neurological complications was determined using the International Statistical Classification of Diseases and Related Health Problems, 10th Revision (ICD-10). Except for the duration of mechanical ventilation, ICU stay, hospitalization, and tumor recurrence or metastasis, all other secondary outcomes were observed within 30 days post-surgery.

Statistical analysis

All statistical analyses were conducted using SPSS 15.0 (IBM, Armonk, NY, USA), and graphical illustrations were created using GraphPad Prism 6 (GraphPad Software Inc., La Jolla, CA, USA). Descriptive statistical methods were employed, including mean ± standard deviation ( x¯ ± SD) for continuous variables or median with interquartile range (IQR, 25th–75th percentiles).

The Shapiro-Wilk test was applied to evaluate the normality of continuous variables. For normally distributed data, the T-test was used for comparisons, while the Mann-Whitney U test was employed for data that did not follow a normal distribution. Pearson’s Chi-square test was utilized to analyze categorical variables.

To account for potential confounding factors, logistic regression analysis was performed to evaluate the association between massive RBC transfusion and clinical outcomes. For each result, we first conducted a bivariate analysis. Any variable with a p-value < 0.2 is considered a candidate variable for the final model. Then gradually construct a multivariate model. For secondary outcomes, Bonferroni correction was applied to adjust the significance threshold, taking into account the impact of multiple comparisons. Long-term survival analyses were conducted using survival curves. Cox proportional hazards model was used to evaluate the impact of baseline features on survival. Statistical significance was defined as a p-value < 0.05.

Results

Of the 350 patients who underwent brain tumor resection between October 2014 and March 2022 at Guangzhou Women and Children’s Medical Center, Guangzhou Medical University. A total of 306 met the inclusion criteria for this study. Among these, 78 patients were categorized into the massive transfusion (MT) group, while 228 were included in the non-massive transfusion (non-MT) group. The study flow chart is illustrated in Fig. 1. Demographic information is summarized in Table 1. While the MT group exhibited a lower average body weight compared to the non-MT group, no significant differences were observed in other demographic characteristics. However, the following variables were imbalanced between the two groups: Karnofsky Performance Scale (KPS) scores, edema presence, tumor size, tumor count, intraoperative fluid administration (crystalloids and colloids), blood loss, surgical duration, Ki67 proliferation index, and WHO tumor grades (p < 0.05, Table 2).

Figure 1 The flow chart of the study cohort.

Table 1 Comparison of the demographic characteristics of study patients.

Variables	Total (N = 306)	Non-MT (N = 228)	MT (N = 78)	p value	
Male (cases (%))	179 (58.5)	130 (57.0)	49 (62.8)	0.369	
Age (months)	5.5 (3.0, 9.0)	5.0 (2.0, 9.0)	6.0 (3.0, 9.0)	0.633	
Body weight (kg)	12.0 (12.0, 15.0)	13.0 (10.0, 16.0)	10. 0 (7.9, 11.6)	0.000**	
Hydrocephalus surgical treatment (cases (%))	53 (17.4)	34 (15.0)	19 (24.4)	0.062	
Anaemia (cases (%))	53 (17.3)	35 (15.4)	18 (23.1)	0.120	
Coagulopathy (cases (%))	20 (6.5)	17 (7.5)	3 (3.9)	0.266	
ASA physical status II/III (cases (%))	53 (77.5)	174 (76.3)	63 (80.8)	0.417	
Notes:

MT, Massive transfusion; KPS, Karnofsky performance scale; ASA, American society of anesthesiologists.

** p < 0.01.

Table 2 Comparison of the perioperative, intraoperative and postoperative characteristics of study patients.

Variables	Total (N = 306)	Non-MT (N = 228)	MT (N = 78)	p value	
Preoperative					
KPS score	80.0 (70.0, 90.0)	80.0 (70.0, 90.0)	80.0 (70.0, 80.0)	0.025*	
Edema (cases (%))	148 (48.4)	96 (42.1)	52 (66.7)	0.000**	
Subtentorial tumor (cases (%))	133 (43.46)	99 (43.42)	34 (43.59)	0.979	
Tumor size (cm)	4.3 (3.0, 5.8)	3.8 (2.3, 5.1)	6.1 (5.0, 7.2)	0.000**	
Tumor number (multiple) (cases (%))	39 (12.83)	22 (9.73)	17 (21.79)	0.006**	
Tumor invasiveness (cases (%))	53 (17.3)	34 (14.9)	19 (24.4)	0.057	
Intraoperative					
Brain tumor resection score
(cases (%))				0.132	
2	202 (71.9)	49 (65.3)	153 (74.3)		
3	16 (5.7)	4 (5.3)	12 (5.8)		
4	42 (15.0)	18 (24.0)	24 (11.7)		
5	20 (7.1)	4 (5.3)	16 (7.8)		
Crystalloids infusion (ml/kg)	73.3 (52.3, 104.6)	61.9 (47.4, 82.3)	126.1 (92.2, 186.6)	0.000**	
Colloids infusion (ml/kg)	21.4 (10.4, 35.0)	18.6 (9.0, 27.9)	43.8 (25.0, 63.5)	0.000**	
Blood loss (ml/kg)	20.0 (9.1, 57.1)	13.3 (6.3, 28.6)	137.4 (70.2, 225.0)	0.000**	
Surgical time (min)	271.0 (220.0, 345.0)	250.0 (205.0, 305.0)	368.5 (280.0, 435.0)	0.000**	
Postoperative					
Ki67-proliferation rate	10.0 (1.0, 40.0)	3.000 (1.0, 30.0)	27.500 (4.3, 67.5)	0.000**	
WHO grade (cases (%))				0.000**	
I	127 (51.5)	114 (50.0)	13 (16.7)		
II	42 (13.7)	29 (12.7)	13 (16.7)		
III	32 (10.5)	14 (6.1)	18 (23.1)		
IV	105 (34.3)	71 (31.1)	34 (43.6)		
Notes:

MT, Massive transfusion; WHO, world health organization.

* p < 0.05.

** p < 0.01.

Effects of massive transfusion on postoperative complications

Short-term clinical outcomes are detailed in Table 3. The MT group demonstrated significantly higher 30-day mortality (26.9% vs. 4.8%, p < 0.001), extended postoperative mechanical ventilation times (67.0 h (40.5–157.1) vs. 21.6 h (7.1–65.8), p < 0.001), prolonged ICU stays (6.0 days (3.3–11.8) vs. 2.5 days (1.0–5.0), p < 0.0010), and longer overall hospitalizations (28.0 days (20.0–36.0) vs. 20.0 days (14.0–26.0), p < 0.001) compared to the non-MT group. Additionally, postoperative intracranial hypertension occurred in significantly more patients in the MT group than in the non-MT group (35.9% vs. 8.3%, p < 0.001).

Table 3 In-hospital complications and complications in the patients massive transfused and non- massive transfused.

Variables	Total (N = 306)	Non-MT (N = 228)	MT (N = 78)	p value	
Ventilation support (h)	33.0 (11.0, 86.0)	21.6 (7.1, 65.8)	67.0 (40.5, 157.1)	0.000**	
ICU stay (d)	3.0 (2.0, 6.0)	2.5 (1.0, 5.0)	6.0 (3.3, 11.8)	0.000**	
Hospital stay (d)	21.0 (15.0, 30.0)	20.0 (14.0, 26.0)	28.0 (20.0, 36.0)	0.000**	
Infection (cases (%))	80 (26.1)	55 (24.1)	25 (32.1)	0.169	
Re-craniotomy (cases (%))	16 (5.2)	14 (6.1)	2 (2.6)	0.352	
Intracranial hypertension (cases (%))	47 (15.4)	19 (8.3)	28 (35.9)	0.000**	
New neurologic events (cases (%))	18 (5.9)	13 (5.73)	5 (6.41)	0.954	
Recurrence or metastasis (cases (%))	16 (5.2)	2 (0.9)	4 (5.1)	0.062	
30-day mortality (cases (%))	32 (10.5)	11 (4.8)	21 (26.9)	0.000**	
Notes:

MT, Massive transfusion; ICU, intensive care unit.

** p < 0.01.

Effects of massive transfusion on primary outcomes

Logistic regression analysis revealed that perioperative massive RBC transfusion was independently associated with increased 30-day mortality after adjusting for confounding variables (OR: 0.137, 95% CI [0.036–0.528], p = 0.004; Table 4). However, no significant association was found between massive transfusion and 1-year survival rates. After adjusting confounding factors using multivariable cox regression analysis, it was found that WHO grade and massive transfusion were associated with long-term survival (Table S2). The survival analysis showed that the 1-year survival rate was 35.9% for the MT group, compared to 57.2% for the non-MT group. Correspondingly, the MT group had a lower long-term survival rate than the non-MT group (HR: 1.000, 95% CI [1.000–1.000], p = 0.029; Fig. 2).

Table 4 Multivariable logistic regression analysis of postoperative 30-day and 1-year mortality rate.

Variables	30-day mortality	1-year survival rate	
OR	95% CI	p value	OR	95% CI	p value	
Preoperative							
Massive transfusion	0.137	[0.036–0.528]	0.004**	0.509	[0.184–1.409]	0.194	
Male vs. female	0.852	[0.302–2.403]	0.761	1.380	[0.736–2.588]	0.315	
Age (months)	0.930	[0.796–1.086]	0.356	1.011	[0.923–1.107]	0.811	
Body weight (kg)	0.888	[0.772–1.021]	0.095	0.968	[0.901–1.040]	0.376	
Hydrocephalus surgical treatment	2.165	[0.702–6.670]	0.179	1.042	[0.441–2.462]	0.926	
KPS score	1.005	[0.953–1.060]	0.850	0.988	[0.957–1.021]	0.471	
Anaemia	1.910	[0.407–8.960]	0.412	0.530	[0.244–1.147]	0.107	
Coagulopathy	3.067	[0.000–null]	0.981	0.544	[0.167–1.770]	0.312	
Edema	2.165	[0.702–6.670]	0.179	0.574	[0.296–1.112]	0.100	
Subtentorial tumor	2.085	[0.722–6.023]	0.175	0.555	[0.291–1.057]	0.073	
Tumor size (cm)	1.010	[0.732–1.394]	0.952	1.163	[0.972–1.393]	0.100	
Tumor number (multiple)	0.224	[0.070–0.718]	0.012**	0.357	[0.131–0.973]	0.044*	
Tumor invasiveness	1.026	[0.287–3.666]	0.969	0.628	[0.279–1.417]	0.263	
Intraoperative							
Brain tumor resection score	1.423	[0.769–2.634]	0.261	0.837	[0.596–1.173]	0.301	
Crystalloids infusion (ml/kg)	0.991	[0.983–1.000]	0.042*	1.002	[0.996–1.007]	0.538	
Colloids infusion (ml/kg)	1.005	[0.985–1.026]	0.634	0.994	[0.982–1.007]	0.395	
Blood loss (ml/kg)	1.000	[0.993–1.007]	0.964	0.996	[0.989–1.003]	0.262	
Surgical time (min)	0.999	[0.993–1.005]	0.816	1.004	[1.000–1.008]	0.080	
Postoperative							
Ki67-proliferation rate	1.008	[0.985–1.032]	0.485	1.002	[0.986–1.018]	0.797	
WHO grade	0.369	[0.195–0.700]	0.002**	0.376	[0.270–0.524]	0.000**	
Notes:

KPS, Karnofsky performance scale; WHO, world health organization.

* p < 0.05.

** p < 0.01.

Figure 2 Survival curves of patients’ survival in the MT and non-MT groups MT, massive transfusion.

Mortality details showed that 21 of 78 patients (26.9%) in the MT group died within 30 days, compared to 11 of 228 patients (4.8%) in the non-MT group. At 1 year, 50 of 78 patients (64.1%) in the MT group had died, compared to 98 of 228 patients (42.8%) in the non-MT group (Fig. 3). Additionally, Fig. 4 compares blood transfusion volumes between survivors and non-survivors at both the 30-day and 1-year time points. At these two time points, the blood transfusion volumes of survivors were not decreased compared to non-survivors (p > 0.05).

Figure 3 Chart depicting massive allogeneic RBC transfusion and patients’ mortality rate.

Figure 4 Box plot comparing allogeneic RBC transfusion of subjects who were survival and who were death.

Effects of massive transfusion on secondary outcomes

The adjusted logistic regression model demonstrated a significant association between massive RBC transfusion and an increased risk of postoperative intracranial hypertension (OR: 4.788, 95% CI [1.547–14.824], p = 0.007; Table 5, Table S3). However, no significant associations were observed between massive transfusion and other complications, including infections, re-craniotomy, new neurologic events, tumor recurrence, or metastasis.

Table 5 Multivariable logistic regression analysis of the association between massive transfusion and secondary outcomes.

Secondary outcomes	OR/ß	95% CI	p value	
Infection	0.693	[0.273–1.761]	0.441	
Re-craniotomy	2.617	[0.050–136.317]	0.633	
Intracranial hypertension	4.788	[1.547–14.824]	0.007**	
New neurologic events	5.312	[0.760–37.124]	0.092	
Recurrence or metastasis	0.103	[0.007–1.422]	0.090	
Ventilation support (h)	0.247	[58.739–147.895]	0.000**	
ICU stay (d)	0.184	[3.874–15.077]	0.001**	
Hospital stay (d)	0.118	[−0.040 to 9.360]	0.053	
Note:

** p < 0.01.

Patients in the MT group were significantly more likely to require prolonged mechanical ventilation (OR: 0.247, 95% CI [58.739–147.895], p < 0.001). Furthermore, logistic regression identified a significant relationship between massive transfusion and longer ICU stays (ß = 0.184, 95% CI [3.874–15.077], p = 0.001; Table S4).

Discussion

This study evaluated the clinical outcomes of perioperative massive red blood cell (RBC) transfusion in pediatric patients undergoing brain tumor resection. Among 306 patients included, 78 (25.5%) received massive RBC transfusion, while 228 (74.5%) did not. Our findings revealed that massive transfusion was significantly associated with increased morbidity of postoperative complications and mortality.

First, multivariate analysis identified perioperative massive RBC transfusion as an independent risk factor for 30-day mortality. Recent studies in neurosurgical management have similarly shown that blood transfusion is independently linked to higher 30-day postoperative mortality rates (Mofor et al., 2022). Although the surgical mortality of pediatric brain tumor patients has declined in recent years, it remains closely tied to hemorrhage and the consequent need for transfusion (Liu et al., 2021). Massive transfusions are associated with cardiovascular instability (Yu, Cohen & Parker, 2015), pulmonary dysfunction (McVey et al., 2019; Cernak et al., 2019), and metabolic disturbances (Hendrickson & Hillyer, 2009). Additionally, complications such as fluid and electrolyte imbalances, hypothermia, and autonomic instability can arise, contributing to vasopressor hyporesponsiveness (Pan et al., 2020; Berkowitz et al., 2022). Perioperative coagulopathy, induced by extensive blood replacement, is another critical complication, with 93% of patients showing abnormal hemostasis tests during massive transfusions (Muirhead & Weiss, 2017). These imbalances often stem from inflammatory pathway activation, resulting in enhanced procoagulation and reduced anticoagulation mechanisms (Andreason & Pohlman, 2016; Pohlman et al., 2015).

Second, survival analysis revealed that 1-year survival rates were lower in the massive transfusion group compared to the non-massive transfusion group (p = 0.029). Box plot analysis further demonstrated a dose-response relationship between transfusion volume and postoperative survival. However, multivariate analysis did not find a direct association between massive transfusion and 1-year survival. This suggests that the adverse effects of massive transfusion primarily impact short-term outcomes, without significantly affecting long-term survival. Factors such as tumor histology, delayed diagnosis, and extent of surgical resection remain the primary determinants of long-term survival in neurosurgical patients (Raimondi & Tomita, 1983).

Third, our findings also showed a significant association between massive RBC transfusion and increased requirements for mechanical ventilation and prolonged ICU stays. These results align with earlier studies, which reported that patients receiving RBC transfusion often experience extended hospital stays due to their more severe medical conditions (Neef et al., 2021; Piastra et al., 2008; Cohen et al., 2017). A study using the National Surgical Quality Improvement Program database also demonstrated similar risks of prolonged hospitalization in transfused cranial surgery patients (Glance et al., 2011). However, we also noticed that regression results of the duration of mechanical ventilation showed wide confidence intervals, and we consider that it may be due to the high variability of the effect variables in the sample. Therefore, although the results showed statistical significance (p < 0.05), further research is also needed to validate them.

Finally, massive transfusion was significantly associated with postoperative intracranial hypertension. Excessive RBC transfusion can exacerbate systemic inflammation, increase tissue permeability, induce volume overload, and disrupt hydrostatic equilibrium (Bosboom et al., 2018; Roubinian et al., 2018; Parmar et al., 2017). Although previous research has linked allogeneic RBC transfusion to heightened risks of postoperative infection, tumor recurrence, and metastasis (Xu et al., 2021), our study did not confirm these associations. Differences in patient populations, surgical practices, and transfusion thresholds may explain this discrepancy, highlighting the need for further research to address these confounding factors.

Limitations

This study has several limitations. Firstly, its retrospective design inherently restricts the ability to establish causation. Secondly, the research was carried out at a single institution, which could restrict the broader applicability of its findings. Thirdly, in our case, the number of patients with certain exclusion criteria (such as preoperative blood transfusion history or recurrent tumors) was very small, and in order to reduce population heterogeneity, these patients were excluded. Therefore, our results are not applicable to these exclusion populations. Finally, the observational nature of the study precludes definitive cause-effect conclusions. To date, no large-scale randomized controlled trials have evaluated the effects of transfusion on clinical outcomes in this context. Such trials, especially those comparing varying transfusion thresholds, are urgently needed to validate these findings.

Additionally, this study used non-leukocyte-depleted RBCs, which differ from the leukocyte-depleted RBCs commonly employed in the USA and parts of Europe. This discrepancy further limits the applicability of our findings in other healthcare settings. Future studies should address these regional differences and incorporate larger, more diverse patient populations.

Conclusion

Children receiving massive perioperative RBC transfusion during brain tumor surgery faced significantly higher risks of 30-day mortality and postoperative intracranial hypertension compared to those not receiving massive transfusion. These findings underscore the critical need to carefully balance the risks and benefits of allogeneic transfusion in pediatric neurosurgical patients.

Supplemental Information

Supplemental Information 1 Raw data.

Supplemental Information 2 Definitions of postoperative new neurologic events.

Supplemental Information 3 Multivariable logistic regression analysis of postoperative secondary outcomes.

Supplemental Information 4 Multivariable logistic regression analysis of postoperative secondary outcomes.

Supplemental Information 5 Multivariable logistic regression analysis of postoperative secondary outcomes.

Additional Information and Declarations

Competing Interests

The authors declare that they have no competing interests.

Author Contributions

Yingyi Xu conceived and designed the experiments, prepared figures and/or tables, and approved the final draft.

Na Zhang conceived and designed the experiments, prepared figures and/or tables, and approved the final draft.

Xinxu Ou performed the experiments, prepared figures and/or tables, and approved the final draft.

Yuyin Ye performed the experiments, prepared figures and/or tables, and approved the final draft.

Jianhua Liu performed the experiments, authored or reviewed drafts of the article, and approved the final draft.

Siyi Zhang analyzed the data, authored or reviewed drafts of the article, and approved the final draft.

Xinke Xu analyzed the data, authored or reviewed drafts of the article, and approved the final draft.

Yu Gao analyzed the data, authored or reviewed drafts of the article, and approved the final draft.

Wenchu Chen analyzed the data, authored or reviewed drafts of the article, and approved the final draft.

Xingrong Song conceived and designed the experiments, authored or reviewed drafts of the article, and approved the final draft.

Human Ethics

The following information was supplied relating to ethical approvals (i.e., approving body and any reference numbers):

This study was approved by Research Ethics Committee of Women and Children’s Medical Center, Guangzhou Medical University (Approval Number: 2024-239A01, Date: July 26, 2024). A retrospective analysis was conducted on pediatric patients who underwent brain tumor resection at Guangzhou Women and Children’s Medical Center, Guangzhou Medical University, between October 2014 and March 2022. Written informed consent was obtained from the patients’ legal guardians, authorizing the use of their data for research purposes.

Data Availability

The following information was supplied regarding data availability:

The raw measurements are available in the Supplemental File.

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
