# Peer review of "Effects of perioperative massive transfusion on postoperative outcomes of children undergoing brain tumor removal: a retrospective cohort study"

_PeerJ, doi:10.7717/peerj.19350_

## Round 0.1 · original submission · Major Revisions

While both reviewers find merit in your work, one reviewer has raised several substantive points that require thorough attention. Therefore, I have decided to recommend a major revision to ensure all concerns are adequately addressed. Please carefully consider all reviewers' comments and provide a detailed point-by-point response with your revised manuscript.

Reviewer 1 ·

Basic reporting

This retrospective cohort study investigates the impact of massive perioperative red blood cell (RBC) transfusion on both short-term and long-term outcomes in pediatric patients undergoing brain tumor resection. The study included 306 patients, with 78 in the massive transfusion group and 228 in the non-massive transfusion group. The primary outcomes assessed were 30-day all-cause mortality and 1-year survival rates. The authors found that massive transfusion was significantly associated with increased 30-day mortality, higher incidences of postoperative intracranial hypertension, prolonged mechanical ventilation, and longer ICU stays. However, there was no significant association between massive transfusion and 1-year survival rates. The authors conclude that careful consideration of the risks and benefits of transfusion is crucial in pediatric brain tumor surgery. However, the following concerns should be addressed.

Experimental design

1.In the multivariable regression analysis, did the authors adequately consider all potential confounding factors? For instance, did they control for physiological indicators (e.g., preoperative anemia, platelet count) or intraoperative factors (e.g., blood loss, surgical duration) that might influence the clinical outcomes of the patients?
2.When selecting variables to include in the regression model, did the authors conduct statistical tests or provide theoretical rationale to ensure the validity of the model?
3.In the survival analysis, the authors chose 30-day mortality and 1-year survival rate as the primary outcome indicators. Why were these two time points selected for survival analysis? Did the authors consider other potential long-term outcomes, such as neurological recovery or quality of life?
4.In constructing survival curves, did the authors account for baseline differences among patients, especially with respect to tumor type, surgical approach, and preoperative health status?
5.The authors applied Bonferroni correction to adjust the significance level for multiple comparisons. Could the authors discuss why they chose Bonferroni correction? Did they consider alternative methods for correction, such as false discovery rate control or the Benjamini-Hochberg procedure?
6.The article defines significance as p < 0.05, but certain regression results exhibit wide confidence intervals (e.g., OR for extended mechanical ventilation: 0.247, 95% CI: 58.739–147.895), suggesting potential statistical instability. How do the authors explain these wide confidence intervals? How does this uncertainty affect the clinical interpretation of the results?
7.In the survival analysis, the authors used 30-day mortality as a short-term outcome and 1-year survival rate as a long-term outcome. However, did the authors consider treating the time to death as a continuous variable for more detailed survival analysis? Would using continuous time to death improve the precision of the analysis?
8.The authors excluded patients with certain conditions, such as a history of preoperative blood transfusion or recurrent tumors. Did the authors consider the potential impact of these exclusion criteria on the results of the study? For example, could these exclusion criteria introduce selection bias that might affect the generalizability of the findings?
9.The authors defined "massive transfusion" as the transfusion of red blood cells equivalent to or exceeding the estimated preoperative blood volume within 48 hours post-surgery. How was this threshold determined? Did the authors consider that patients of different age groups or body weights might require different transfusion volumes to achieve the same clinical effect?

Validity of the findings

10.The study mentioned the use of the Karnofsky Performance Scale (KPS) and other preoperative health status assessments. Did the authors account for other factors that might affect clinical outcomes, such as comorbidities (e.g., cardiovascular disease, diabetes)? How might these factors influence the study’s findings?
11.The study design mentions the recording of various intraoperative variables such as surgical duration, blood loss, and type of surgery. How did the authors ensure that these factors were balanced across the groups? Could these intraoperative variables have a significant impact on the results?
12.The study compares patients who received massive transfusion with those who did not. Did the authors account for potential differences between these groups, particularly in terms of tumor staging, tumor type, or preoperative treatments? Did they analyze the potential impact of these differences on the results?

Additional comments

This retrospective cohort study investigates the impact of massive perioperative red blood cell (RBC) transfusion on both short-term and long-term outcomes in pediatric patients undergoing brain tumor resection. The study included 306 patients, with 78 in the massive transfusion group and 228 in the non-massive transfusion group. The primary outcomes assessed were 30-day all-cause mortality and 1-year survival rates. The authors found that massive transfusion was significantly associated with increased 30-day mortality, higher incidences of postoperative intracranial hypertension, prolonged mechanical ventilation, and longer ICU stays. However, there was no significant association between massive transfusion and 1-year survival rates. The authors conclude that careful consideration of the risks and benefits of transfusion is crucial in pediatric brain tumor surgery. However, the following concerns should be addressed.
1.In the multivariable regression analysis, did the authors adequately consider all potential confounding factors? For instance, did they control for physiological indicators (e.g., preoperative anemia, platelet count) or intraoperative factors (e.g., blood loss, surgical duration) that might influence the clinical outcomes of the patients?
2.When selecting variables to include in the regression model, did the authors conduct statistical tests or provide theoretical rationale to ensure the validity of the model?
3.In the survival analysis, the authors chose 30-day mortality and 1-year survival rate as the primary outcome indicators. Why were these two time points selected for survival analysis? Did the authors consider other potential long-term outcomes, such as neurological recovery or quality of life?
4.In constructing survival curves, did the authors account for baseline differences among patients, especially with respect to tumor type, surgical approach, and preoperative health status?
5.The authors applied Bonferroni correction to adjust the significance level for multiple comparisons. Could the authors discuss why they chose Bonferroni correction? Did they consider alternative methods for correction, such as false discovery rate control or the Benjamini-Hochberg procedure?
6.The article defines significance as p < 0.05, but certain regression results exhibit wide confidence intervals (e.g., OR for extended mechanical ventilation: 0.247, 95% CI: 58.739–147.895), suggesting potential statistical instability. How do the authors explain these wide confidence intervals? How does this uncertainty affect the clinical interpretation of the results?
7.In the survival analysis, the authors used 30-day mortality as a short-term outcome and 1-year survival rate as a long-term outcome. However, did the authors consider treating the time to death as a continuous variable for more detailed survival analysis? Would using continuous time to death improve the precision of the analysis?
8.The authors excluded patients with certain conditions, such as a history of preoperative blood transfusion or recurrent tumors. Did the authors consider the potential impact of these exclusion criteria on the results of the study? For example, could these exclusion criteria introduce selection bias that might affect the generalizability of the findings?
9.The authors defined "massive transfusion" as the transfusion of red blood cells equivalent to or exceeding the estimated preoperative blood volume within 48 hours post-surgery. How was this threshold determined? Did the authors consider that patients of different age groups or body weights might require different transfusion volumes to achieve the same clinical effect?
10.The study mentioned the use of the Karnofsky Performance Scale (KPS) and other preoperative health status assessments. Did the authors account for other factors that might affect clinical outcomes, such as comorbidities (e.g., cardiovascular disease, diabetes)? How might these factors influence the study’s findings?
11.The study design mentions the recording of various intraoperative variables such as surgical duration, blood loss, and type of surgery. How did the authors ensure that these factors were balanced across the groups? Could these intraoperative variables have a significant impact on the results?
12.The study compares patients who received massive transfusion with those who did not. Did the authors account for potential differences between these groups, particularly in terms of tumor staging, tumor type, or preoperative treatments? Did they analyze the potential impact of these differences on the results?

Reviewer 2 ·

Basic reporting

This study retrospectively analyzed the effect of perioperative massive transfusion on long-term outcomes of children undergoing brain tumor removal. The authors demonstrated that massive transfusion has a pronounced impact on short-term outcomes, particularly increasing perioperative mortality and complication risks in children undergoing brain tumor surgery. Brain tumor poses a serious threat to children’s health and lives worldwide. Perioperative blood transfusion is very common in children with brain tumor removal. These findings have important implications for balancing the risks and benefits of allogeneic transfusion in pediatric neurosurgical patients. However, some comments may be addressed to improve the manuscript.
1. The authors should definite the use of “short-term” and “long-term” in the Title and Abstract, as well as in the main text. “Short-term” outcomes included 30-day all-cause mortality and postoperative complications? The “long-term” outcome means 1-year survival rates? The Result section of Abstract only involves “short-term” outcomes. There was no description concerning “1-year survival rates” in the Result section of Abstract.
2. The authors should describe in more detail about the influences of perioperative allogeneic transfusion in patients with brain tumor resection in the Introduction section. Besides, it should not be limited only on “risks”, but also on “benefits”.
3. The End time of data collection should be mentioned in the Study Design section.
4. Table 2 is not quoted, please check.
5. In the Table 3, “The MT group demonstrated significantly higher 30-day mortality (26.9% vs. 4.8%)”. However, in the Fig 3. “Mortality details showed that 22 of 78 patients (28.2%) in the MT group died within 30 days, compared to 11 of 228 patients (4.8%) in the non-MT group.” (Line . The author should double-check this data.
6. The description concerning Fig 4 should be extended.
7. The Line 182, “Patients in the MT group were significantly more likely to require prolonged mechanical ventilation”. Patients in the MT group had longer mechanical ventilation time, which is a result we observed. The use of “more likely” is inappropriate.
8. What is “morbidity” in the line 191.
9. For patient with brain tumor resection, perioperative blood transfusion has great benefits for reducing cerebral ischemia and anemia. It is important for balancing the risks and benefits of allogeneic transfusion in the clinic. The authors should revise the Discussion section.
10. Fig 1. The authors should exhibit the groups of massive transfusion and non-massive transfusion.
11. The overall language of the article is good. But there are still a few details that need attention. Such as, “outcomes” , The use of “30-day mortality”, “thirty-day mortality”, “thirty-day all-cause mortality” should be unification in the manuscript. The author should check the full text more carefully.
12. The authors should further conduct subgroup analysis based on tumor size, number, intraoperative blood loss, ki67 - proliferation rate, WHO grade, etc.

Experimental design

Comments are merged into Basic reporting

Validity of the findings

Comments are merged into Basic reporting

---

## Round 0.2 · accepted · Accept

The revisions have adequately addressed the previous concerns raised by both reviewers. Based on the positive feedback from both reviewers and your careful revisions, I am pleased to inform you that your manuscript has been accepted for publication.

Reviewer 1 ·

Basic reporting

All issues in the revised draft have been addressed. It is recommended to accept it

Experimental design

The design has no issues

Validity of the findings

the findings has significance

Additional comments

no

Reviewer 2 ·

Basic reporting

The author's revised manuscript has been effectively improved.

Experimental design

no comment

Validity of the findings

no comment